# Comparison of Hodgkin’s Lymphoma in Children and Adolescents. A Twenty Year Experience with MH’96 and LH2004 AIEOP (Italian Association of Pediatric Hematology and Oncology) Protocols

**DOI:** 10.3390/cancers12061620

**Published:** 2020-06-18

**Authors:** Roberta Burnelli, Giulia Fiumana, Roberto Rondelli, Marta Pillon, Alessandra Sala, Alberto Garaventa, Emanuele S.G. D’Amore, Elena Sabattini, Salvatore Buffardi, Maurizio Bianchi, Luciana Vinti, Marco Zecca, Paola Muggeo, Massimo Provenzi, Piero Farruggia, Francesca Rossi, Salvatore D’Amico, Elena Facchini, Sayla Bernasconi, Raffaela De Santis, Tommaso Casini, Fulvio Porta, Irene D’Alba, Rosamaria Mura, Federico Verzegnassi, Antonella Sau, Simone Cesaro, Katia Perruccio, Monica Cellini, Patrizia Bertolini, Domenico Sperlì, Roberta Pericoli, Daniela Galimberti, Adele Civino, Maurizio Mascarin

**Affiliations:** 1Pediatric Hemato-Oncology Unit, Azienda Ospedaliero Universitaria Sant’Anna di Ferrara, 44124 Cona, Ferrara, Italy; 2Department of Pediatrics, University of Modena and Reggio Emilia, Azienda Ospedaliero-Universitaria Policlinico, 41121 Modena, Italy; 273790@studenti.unimore.it (G.F.); cellini.monica@aou.mo.it (M.C.); 3Department of Pediatrics, Sant’Orsola Hospital, University of Bologna, 40138 Bologna, Italy; roberto.rondelli@aosp.bo.it (R.R.); elena.facchini@aosp.bo.it (E.F.); 4Pediatric Hematology, Oncology and Stem Cell Transplant Center, Department of Women and Child’s Health, University of Padua, 35122 Padova PD, Italy; marta.pillon@unipd.it; 5Department of Paediatrics, Ospedale San Gerardo, University of Milano-Bicocca, Fondazione MBBM, 20900 Monza, Italy; ale.sala@asst-monza.it; 6Department of Pediatric Oncology, IRCCS Istituto Giannina Gaslini, 16147 Genoa, Italy; albertogaraventa@gaslini.org; 7Department of Pathological Anatomy, San Bortolo Hospital, 36100 Vicenza, Italy; emanuele.damore@gmail.com; 8Department of Experimental, Diagnostic and Speciality Medicine, University of Bologna, 40126 Bologna, Italy; elena.sabattini@aosp.bo.it; 9Paediatric Haemato-Oncology Department, Santobono-Pausilipon Children’s Hospital, 80122 Napoli, Italy; salvatorebuffardi@hotmail.it; 10Pediatric Onco-Hematology and Stem Cell Transplant Division, City of Health and Science, Regina Margherita Children’s Hospital, 10126 Turin, Italy; maurizio.bianchi@unito.it; 11Department of Pediatric Hematology and Oncology, IRCSS Ospedale Bambino Gesù, 00165 Rome, Italy; luciana.vinti@opbg.net; 12Oncoematologia Pediatrica, Fondazione IRCCS Policlinico San Matteo, 27100 Pavia, Italy; m.zecca@smatteo.pv.it; 13Department of Pediatric Oncology and Hematology, University Hospital of Policlinico, 70124 Bari, Italy; paola.muggeo@gmail.com; 14Department of Pediatrics, Civic Hospital, 24127 Bergamo, Italy; mprovenzi@asst-pg23.it; 15Pediatric Hematology and Oncology Unit, Oncology Department, A.R.N.A.S. Ospedali Civico, Di Cristina e Benfratelli, 90127 Palermo, Italy; piero.farruggia@arnascivico.it; 16Department of Woman, Child and of General and Specialist Surgery, Second University of Naples, 81100 Naples, Italy; francesca.rossi@unina2.it; 17Department of Clinical and Experimental Medicine, Paediatric Haemato-Oncology Unit, University of Catania, 95124 Catania, Italy; sdamico@unict.it; 18Pediatric Hematology Oncology, Bone Marrow Transplant, S. Chiara University Hospital of Pisa, 56126 Pisa, Italy; s.bernasconi@ao-pisa.toscana.it; 19IRCCS Casa Sollievo della Sofferenza, San Giovanni Rotondo, 47156 Foggia, Italy; r.desantis@operapadrepio.it; 20Paediatric Haematology-Oncology Unit, Meyer Paediatric Hospital, 50139 Florence, Italy; tommaso.casini@meyer.it; 21Oncology-Hematology and BMT Unit, Ospedale dei Bambini, Spedali Civili, 25123 Brescia, Italy; fulvio.porta@gmail.com; 22Division of Pediatric Hematology and Oncology, Ospedale G. Salesi, 60123 Ancona, Italy; irene.dalba@ospedaliriuniti.marche.it; 23Pediatric Hematology and Oncology Unit, Ospedale Pediatrico Microcitemico, 09121 Cagliari, Italy; rosamaria.mura@aob.it; 24Paediatric Onco-Haematology Unit, “Burlo Garofolo” Hospital, 34137 Trieste, Italy; federico.verzegnassi@burlo.trieste.it; 25Pediatric Hematology-Oncology Unit, Ospedale Civico, Pescara, Italy; antonella.sau1@gmail.com; 26Pediatric Hematology Oncology Unit, Department of Mother and Child, Azienda Ospedaliera Universitaria Integrata, 37126 Verona, Italy; simone.cesaro@aovr.veneto.it; 27Pediatric Hematology-Oncology Unit, S. Maria della Misericordia Hospital, 65124 Perugia, Italy; katia.perruccio@ospedale.perugia.it; 28Pediatric Hematology Oncology Unit, Azienda Ospedaliero Universitaria of Parma, 43126 Parma, Italy; PBertolini@ao.pr.it; 29Pediatric Unit, Azienda Ospedaliera Annunziata, 87100 Cosenza, Italy; d.sperli59@gmail.com; 30Pediatric Oncology Unit, Infermi Hospital, 47923 Rimini, Italy; roberta.pericoli@auslrn.net; 31Azienda Ospedaliero Universitaria Senese Policlinico “Le Scotte”, Clinica Pediatrica, 53100 Siena, Italy; d.galimberti@ao-siena.toscana.it; 32Pediatric Rheumatology and Immunology, PO “Vito Fazzi”, 73100 Lecce, Italy; adelecivino@gmail.com; 33AYA Oncology and Pediatric Radiotherapy Unit, CRO-Centro di Riferimento Oncologico di Aviano, IRCCS, 33081 Aviano, Italy; mascarin@cro.it

**Keywords:** Hodgkin’s lymphoma, adolescent, children, pediatric, clinical characteristics, prognosis, chemotherapy, radiotherapy, histology

## Abstract

Adolescents and young adults (AYAs) represent a distinct group of patients. The objectives of this study were: To compare adolescent prognosis to that of younger children; to compare the results achieved with the two consecutive protocols in both age groups; to analyze clinical characteristics of children and adolescents. Between 1996 and 2017, 1759 patients aged <18 years were evaluable for the study. Five hundred and sixty patients were treated with the MH’96 protocol and 1199 with the LH2004 protocol. Four hundred and eighty-two were adolescents aged ≥15 years. Patients in both age groups showed very favorable prognoses. In particular, OS improved with the LH2004 protocol, especially in the adolescent group and in the low risk group, where radiation therapy was spared. Adolescent characteristics differed significantly from the children’s according to sex, histology, and the presence of symptoms. Remarkable is the decrease both in mixed cellularity in the children and in low stages in both age groups in the LH2004 protocol with respect to MH’96 protocol. Based on our experience, adopting pediatric protocols for AYA does not compromise patient outcomes.

## 1. Introduction

Hodgkin’s lymphoma (HL) is the most common cancer diagnosed in adolescents and young adults (AYAs) between 15 and 24 years [1,2]. HLs are responsible for 16% of annual cancer diagnoses in AYAs, and studies suggest that lymphoma-related mortality is higher in AYAs than in younger children [3]. In fact, despite excellent outcomes registered, this age group recently gained attention due to SEER (Surveillance, Epidemiology, and End Results) data evidence of a lack of improvement in survival rates compared to both children and adults [4]. As it is in children, in this age group the major challenge is the balance between radiotherapy and chemotherapy and the general role and need for radiotherapy, especially in early stage disease. Delayed diagnosis, the relative lack of participation of these patients in clinical trials, decreased treatment adherence, and social environment have been pointed out as contributing factors, but no clear etiology has been defined [5,6,7]. It is also noteworthy that the incidence rate of HL in AYAs in Italy (64.6 cases per million) is the highest in Europe (29.7 per million) [8,9]. Research on the characteristics of this age group could help identify specific biological features of the disease, which, in turn, would help define optimal treatment.

Our aims for this study were:To evaluate the prognosis of adolescents in comparison with younger children when treated with the same protocol;to compare the results achieved in both age categories with the most recent Italian Association of Pediatric Hematology and Oncology (AIEOP) chemo-radiotherapeutic protocol (LH2004) and the previous one (MH’96); andto identify significant differences in clinical presentation of children and of adolescents, as well as an analysis of age as a prognostic factor that might impact the outcome of these categories of patients differently.

## 2. Materials and Methods

This particular analysis compares patients aged between 15 and 18 years to those younger than 15 years. All patients were registered at the 40 Italian Pediatric Onco-Hematology Centers applying the MH’96 AIEOP protocol from February 1996 to May 2004 and at the 35 Italian Pediatric Onco-Hematology Centers applying the LH2004 protocol from June 2004 to June 2017 (Appendix A). The results of the MH’96 AIEOP Protocol have already been published, without a specific analysis on adolescents [10].

Diagnosis was histologically confirmed and centrally reviewed according to the WHO classification [11] by the same two pathologists (E.S.G.D.A, E.S.). Patients had a complete anamnesis, and underwent physical examination, routine laboratory tests, chest X-rays, abdomen ultrasound, a neck/chest/abdomen/pelvis contrast-enhanced computed tomography (CT) scan, and were staged according to the Ann Arbor classification. Bone marrow biopsy was performed in patients with stages III-IV or B symptoms. Mediastinal masses were evaluated at diagnosis and after induction therapy with ^67^Ga scan, progressively replaced with ^18^FDG-PET TC. Radiological surveillance after stop therapy included chest X-rays and abdomen ultrasound, and CT scan if necessary. ^18^FDG-PET TC was not suggested for screening during follow-up.

Patients were divided into three therapeutic groups, which were equivalent in both protocols. Group 1 (GR1) included stages IA and IIA with no mediastinal involvement or with mediastinal-thoracic ratio (M/T) < 0.33, less than four nodal regions, and no lung hilar adenopathy. Group 3 (GR3) included stages IIIB and IV, and patients with M/T ≥ 0.33, irrespective of the stage. Group 2 (GR2) consisted of patients not included in GR1 and GR3.

The “Children” group included patients aged <15 years and the “Adolescents” group those between 15 and 18 years.

### 2.1. Response Criteria

Response to treatment was evaluated according to the three therapeutic group’s schedules. Complete response (CR) was defined as the absence of clinical, radiological (ultrasound and CT scan evaluation) and radio-isotopic evidence of disease. Bulky mediastinal involvement was considered in CR with a reduction of ≥75% of the volume (greatest diameter × height × 0.52) and negative Ga scan or 18FDG-PET. Partial response (PR) was defined as tumor volumetric reduction ≥75%, <75%, >50%, ≤50%. Progressive disease (PD) was defined as disease progression during first-line chemotherapy or only transient response (CR or PR) during therapy or within three months from stop therapy. Criteria were either an increase in tumor size in previously involved sites and/or involvement of a new site. Relapse was defined as a pathologically confirmed recurrence of HL after three months from stop therapy [10].

### 2.2. Treatment

#### 2.2.1. MH’96 Protocol

As previously published, GR1 patients received three ABVD. Patients with PR ≤ 50% after the first two courses of chemotherapy were shifted to intensified therapy with two IEP/OPPA/COPP. GR2 and GR3 patients received four and six COPP/ABV (cyclophosphamide, vincristine, procarbazine, prednisolone, adriamycin, bleomycin, vinblastine) respectively at 28-day intervals. Patients with PR ≤ 50% after the first two courses of chemotherapy were shifted to intensified therapy with IEP/OPPA/IEP/OPPA/IEP. Radiotherapy (RT) started four weeks after the completion of chemotherapy with involved fields (IF) RT, defined on anatomical boundaries, that include the entire lymph node regions containing the affected nodes. GR1 patients without initial mediastinal involvement in CR after the end of chemotherapy were not irradiated. All other patients in CR or PR ≥ 75% received low-dose irradiation (20 Gy at 1.8 Gy/fraction), while patients with PR < 75%, got a higher dose (36 Gy). The treatment exceeding 20 Gy was confined to residual mass. Patients with pulmonary or renal involvement persisting after the first two courses received 10 Gy on residual lesions. The liver dose was 12 Gy, and a boost dose up to 15 Gy was given to persistent foci at the end of chemotherapy (Table 1) [10].

#### 2.2.2. LH2004 Protocol

GR1 patients received three ABVD, and no RT in CR patients or 25.2 Gy in PR patients. Patients with PR ≤ 50% after the first two courses of chemotherapy were treated with IEP/OPPA/COPP/IEP and RT (14.4 Gy if CR, 25.2Gy if PR ≥ 50%).

GR2 patients received four COPP/ABV, followed by 14.4 Gy RT in CR patients. Patients in PR after the end of chemotherapy received two further cycles of IEP followed by RT (14.4 Gy/25.2 Gy). Patients with PR ≤ 50% after the first two courses of chemotherapy were treated with IEP/OPPA/IEP/OPPA/IEP and RT (14.4 Gy/25.2 Gy).

GR 3 patients received six COPP/ABV followed by RT (14.4 Gy/25.2 Gy). Patients in PR after four cycles of chemotherapy received two cycles of IEP followed by 14.4 Gy IFRT if CR or two more COPP/ABV if PR, followed by RT (14.4 Gy/25.2 Gy). Patients with PR ≤ 50% after the first two courses of chemotherapy were treated with IEP/OPPA/IEP/OPPA/IEP and RT (14.4 Gy/25.2 Gy). RT started four weeks after the completion of chemotherapy with local fields, including the only lymph node affected area, and not the entire region. Partial responder patients received a boost up to 35 Gy to the residual volume larger than 50 cm^3^ (Table 1).

As in the previous MH’96 protocol, patients with nodular lymphocyte predominance (nLP) histological subtype were treated like patients with common HL (cHL) until 2011, when some centers chose to participate in the European International protocol for LP, named EuroNet-PHL-LP1 (EudraCT 2007-004092-19) for stages I-IIA.

The study was approved by the ethics committee or the institutional review board of each participating institution (Appendix A). Written informed consent was obtained from the parents or legal guardians of all patients. After completion of treatment, biannual monitoring was scheduled for at least five years. Patients’ follow-up was updated in September 2019.

### 2.3. Statistical Methods

All data were collected in a central database (AIEOP MH) and analyzed. Overall survival (OS) was calculated from the date of diagnosis to the date of either death due to any cause or last contact. Event free survival (EFS) is considered to be from the date of diagnosis to the date of either a first event (relapse, second malignant neoplasm—SMN or death), or the last contact for those who are still alive and free of disease. Freedom from progression (FFP) was defined as the interval from the date of diagnosis to that of relapse or PD, or last follow-up for patients without recurrent disease. SMN and death before recurrence were considered competing for events. The Kaplan–Meier method was used to estimate EFS, FFP, and OS probabilities [12], while differences between the groups were calculated using the log-rank test [13]. Results were expressed as a probability (%) and standard error (SE). All *p* values were two-sided, and values <0.05 were considered to be statistically significant.

## 3. Results

The MH’96 protocol enrolled 605 patients: 45 patients were excluded from the analysis because of previous therapy (8), age >18 years (7), lack of information on treatment response (6), or no data beyond registration (24), resulting in 560 patients (92.5%) considered evaluable.

The LH2004 protocol enrolled 1300 patients: 101 were excluded from the analysis because of previous therapy (6), age >18 years (15), wrong diagnosis (5), protocol error (3), immunodeficiency (2), lack of information on treatment response (48), or no data beyond registration (22), resulting in 1199 patients (92.2%) considered evaluable.

Over the 22 years between 1996 and 2017, the 1759 patients <18 years were enrolled and were considered evaluable for the study. There were 482 adolescents aged ≥15 years, representing the 27.4% of the total. Their recruitment progressively grew from 11 per year between 1997 and 1999 to 40 per year in 2015, representing 16.1% of MH’96 patients and the 32.9% of LH2004 patients (*p* = 0.000) (Figure 1 and Table 2).

### 3.1. Patients Characteristics

The demographic and clinical characteristics of analyzed patients are reported in Table 2.

The relative frequency of patient characteristics in the two protocols (Table 2) and in the age categories (Table 3) were compared.

The characteristics of the 482 adolescents analyzed were fairly uniform throughout the two decades, except for stage distribution and the more common extra-nodal disease in the recent study (Table 2). However, the adolescents’ characteristics differed significantly from the children’s according to sex, histology, and presence of symptoms in both protocols (Table 1). In the LH2004 protocol, the male-to-female ratio was inverted in children (M 58.2%, F 41.8%, M/F: 1.4) with respect to adolescents (M 48.2%, F 51.8%, M/F: 0.93). In both protocols nodular sclerosis was more frequent in adolescents than in children (80% and 64.7% vs. 80.9% and 73%), while both mixed cellularity (13.3% and 22.6% vs. 6.4% and 9.8%) and nodular lymphocyte predominance were less frequent (3.3% and 10.9% vs. 4.0% and 9.0%). An important decrease in mixed cellularity frequency in children through the two protocols (MH’96: 22.6%, LH2004: 9.8%) was observed, with a relative increase in nodular sclerosis (MH’96: 64.7%, LH2004: 73.0%). Adolescent patients presented with B-symptoms more frequently than children. Both children and adolescents in the LH2004 protocol were included in higher therapeutic groups than in MH’96.

### 3.2. Survival

The probabilities of OS, FFP, and EFS at 5 and 10 years registered in children and adolescents treated with MH’96 and LH 2004 protocols are reported in Figure 2 and Figure 3. Median observation time was 11.0 years (4 months–17.7 years) and 7.3 years (3 months–15.8 years) respectively.

There was no statistically significant difference in the outcomes of both groups.

The historical comparison between the two protocols showed an improvement of OS with the most recent one (LH2004), especially in low risk patients and in the adolescent group (Table 4). At the same time, EFS and FFP rates were superimposable in both the studies, but with smaller RT volumes and lower doses in the LH2004 protocol.

Patients with nLP HL presented a 10-year OS = 100% in both protocols [10] (Appendix A). In any case, the histotype was not associated with any favorable impact on EFS, neither in the MH’96 protocol [10] nor in the LH2004 (Appendix A), and there was no statistical difference between children and adolescents with nPL.

Second malignant neoplasms were the only long-term side effects considered in this analysis. In the MH’96 protocol 18 SMNs were solid tumors (thyroid carcinoma: 14, breast cancer: 1, lung cancer: 1, leiomyosarcoma: 1, osteosarcoma: 1), while 5 were hematological malignancies (acute myeloblastic leukemia AML: 2, non-Hodgkin lymphoma NHL: 2, mycosis fungoides: 1). A third malignancy (soft tissue sarcoma) occurred after AML. Eighteen were registered in the children group and 5 in the adolescent group. All but one case of solid tumors occurred within the irradiation field after a median time of 7.9 years (2.3–10.2 years). Hematological malignancies occurred at a median time of 4.2 years (11 months–7.4 years) [10]. In the LH2004 protocol, 13 SMNs were solid tumors (thyroid carcinoma: 10, mucoepidermoid carcinoma: 2, Ewing’s sarcoma: 1) and 5 were primary mediastinal large B-cell lymphomas (PMLBCL). Fourteen were diagnosed in the children group and 4 in the adolescent group. Solid tumors occurred at a median time of 6.8 years (4.3–11.6 years). The PMLBCL occurred at a median time of 12.6 months (4–22 months)

## 4. Discussion

In Italy, before 1996 the upper age limit for inclusion in a pediatric HL protocol was 15 years. This was extended up to 18 years for the MH’96 and LH2004 protocols. This study, therefore, includes all adolescents treated in an Italian pediatric protocol from 1996 to 2016, when Italy entered the international EuroNet-PHL-C2 protocol. Comparison with other studies which include older patients up to 21–25 years old might be limited, but those were eliminated by our analysis because of the eligibility, even though treated in some centers. Over the 20 years, adolescent enrollment in AIEOP centers progressively grew from 16.1% of MH’96 patients to 32.9% of LH2004 patients (*p* = 0.000). This reflected a similar increase reported in AYAs with lymphoma and other cancers treated with AIEOP protocols in Italy in the same time span [14,15,16]. This trend was also seen in North America, with 16–21 AYAs: from 28% in POG-9426 between 1996 and 2005, to 38% in COG-AHOD0431 between 2006 and 2009 [17]. Moreover, a recent study from the Dutch pediatric centers recorded an even greater increase in registration of adolescents, from 27% in 2004 to 81% in 2015 [18]. This evidence alone does not suggest an increased incidence of the disease in the overall population, yet it does indicate the progressive referral of adolescents to pediatrics centers.

The characteristics of the 482 adolescents analyzed were uniform throughout the two decades, but differed from those of children in sex, histology, and clinical presentation.

It is well known that the male gender is related to most cancers, and it is strongly associated with each type of lymphoma, including HL. This suggests a role of gender itself in hematologic tumors connected to genetic and immune-related factors, rather than hormonal variations, which better relate to bone tumors and germ-cell tumors (GCTs) [19]. Our patient population is predominantly male, except for adolescents in the most recent protocol, where the male-to-female ratio showed an increase in girls with age, similarly to other publications [18,20,21,22,23].

Histologic subtype distributions present modifications according to age: nodular sclerosis was more frequent in adolescents than in children, while mixed cellularity and lymphocyte predominance was less frequent. This trend registered in our experience was also recorded by other recent researches [22,23,24,25,26]. A peculiarity that emerged in our study was the critical decrease in mixed cellularity frequency in children in the second protocol (22.6% vs. 9.8%). Conversely, there was an increase in nodular sclerosis and not otherwise specified (NOS) cHL, as written in the report on HL incidence in Europe in 1978–1997 [21]. A cause for this variation is not very easily identified. The slightly superior mean and median age in the more recent protocol might be a hypothesis. As for the increase in cHL-NOS, we would suggest two possible explanations. The first one could relate to the long span of time considered in this case series, and the subsequent changes in the available immunohistochemical markers and in the general approach to the diagnosis. Secondly, the increasing use of a needle-biopsy approach in recent decades has certainly limited the precise definition of histological subcategories in daily practice in this diagnostic field, as in many others [27].This could be investigated in future, as children have classically been reported with an incidence of mixed cellularity >30% [28]. This histological type has frequently been associated with Epstein–Barr virus [29], although the infection is more typical of adolescents. In addition, it has been connected to different immune statuses in different age groups [30].

B-symptoms were more frequent among adolescents than children [22]. This characteristic also emerged when comparing adolescents with adults [31], which suggests a direct connection with age rather than with disease stage, which was not more advanced in adolescents than in children.

Another peculiarity of our study, which lasted over an extended period, is that in the LH2004 protocol both children and adolescents were less frequently in low-stage disease, in contrast with other reports [18,22,25], while there was no difference between children and adolescents in stage distribution. Extra-nodal involvement increased, but only in adolescents enrolled in the LH2004 protocol. This trend can be explained by the introduction of PET-CT in clinical practice, with its high sensitivity in detecting active disease [23,32], when compared to CT alone and bone marrow biopsy [33]. Therefore, patients in the LH2004 protocol were less frequently placed in GR1. This could mean that patients with a more recent diagnosis underwent better staging and more appropriate treatment. The use of PET-CT for evaluating response to therapy allows for better identification of residual masses after chemotherapy, while neither CT nor MRI can easily distinguish the fibrotic or vital nature of residual tissues [34]. The use of this response-adapted strategy allows for a de-escalation of treatment intensity, with a reduced dose of radiotherapy, even in patients initially placed in higher-risk groups. We calculated that radiotherapy in LH2004 was avoided in 70% of the GR1 patients compared to the 57% in MH’96, (where only those without an initial mediastinal involvement were considered). In LH2004, 77% of GR2 and 67% of GR3 patients received a low-dose (14.4Gy) irradiation, while 71% of GR2 and only 62% of GR3 patients in the MH’96 trial were given the same.

HL survival progressively increased in the last century, from 10% to >85% 5-year survival rate in all ages since the 1960s. In addition, HL is one of the most curable malignancies nowadays [35]. Pediatric patients have shown highly favorable prognoses in recent reports, which is confirmed in our study for both children and adolescents. In particular, the older patients presented an increased survival rate in the LH2004 protocol (94.9% at 5 years and 93.6% at 10 years) compared to the MH’96 protocol (92.1% at 5 years and 89.9% at 10 years), whereas there was not this difference among the children. This improvement is more evident in low risk patients, where radiation therapy was spared, but is also present in the other therapeutic groups where doses and radiation fields were reduced. These results may reflect better treatment group stratification and tailored therapy due to the introduction of PET-CT.

AYAs gained priority attention following evidence of worse survival in comparison with both children and adults. Adult protocols were utilized in many international studies with different results. Adolescents aged 15–18 treated in the United Kingdom between 1969 and 1998 years showed OS rates lower than those of young adults, aged 20–25 [36,37]. It is not very easy to make a comparison with these studies, because of the old therapeutic regimens applied, together with old staging systems such as laparotomy and splenectomy, which are no longer considered appropriated [38]. In the period 1978–2003, a Greek study showed a progressive improvement in the OS of the 16–23-year-old AYAs treated with three consecutive adult chemotherapeutic regimens [39] and the more recent studies performed between 1979–2013 in Europe and Canada registered 5-year OS rates even higher than 90% [23,40,41]. However, AYA groups have been treated in both pediatric and adult centers without specific criteria, and current research aims to identify the best strategies for this unique group through several studies. Hungarian research between 1990 and 2004 showed less favorable results in 14–18-year-old patients treated with adult protocols, compared to patients aged >18-years and to adolescents treated in pediatric centers [42]. Similarly, a North American study between 1999 and 2006 showed the OS of patients aged 17–21 treated with an adult trial was lower than the OS of young adults (aged 22–44) enrolled in the same trial. It was also inferior to the OS of adolescents aged 17–21, treated with the pediatric COG protocol in 2002–2009 [43] (Table 5).

Unfortunately, there is no published data regarding adolescents treated with adult protocols in Italy, but recent studies conducted elsewhere in Europe showed no significant differences between the two approaches, even if a widespread use of RT and anthracyclines is generally observed in adult trials [24,44]. Data from the Canadian IMPACT (Initiative to Maximize Progress in Adolescent and Young Adult Cancer) study, comparing the outcome of 15–21 year-old AYAs treated in pediatric or adult centers between 1992 and 2012, showed no differences in OS and EFS. Patients treated in the pediatric centers received more frequent radiation treatment, but a lower dose, than adults and showed a higher rate of SMNs. In adult centers, patients received greater cumulative doses of anthracyclines and a higher rate of cardiovascular events was registered, even though not statistically significant [45]. A similar use of RT with more frequency and lower dose in pediatric centers compared to adults was reported in USA [43] (Table 5).

Studies in Europe [18,20,22,42,46,47,48,49,50] and North America [23,43,45], evaluating AYAs included in pediatric protocols show 5-year OS higher than 90%, with no differences compared to children. On the other hand, EFS was lower in adolescents aged >13 years than in younger patients included in the POG 8725 protocol [51]. Similarly, the 13–18-year-old patients registered in the GPOH-HD-2002 protocol showed lower EFS than children [48]. The same disadvantage was observed in adolescents treated in northern Tunisia, who showed a lower EFS than young adults [52]. However, our study, carried out in pediatric oncology centers, showed no significant differences between the two age groups in OS, EFS, and FFP rates, as reported in other studies [18,20,23,46,47,48,50] (Table 5).

Given the very favorable survival rates achieved in the last decade in AYAs affected by HL, future studies should focus not only on treatment efficacy but also on identifying the best approaches, in terms of a better quality of life during therapy and reduced long-term negative side effects [53,54,55,56]. The choice of OS as the only parameter defining outcome excludes relapses or long-term harmful effects that profoundly affect the quality of life [57]. One useful end-point could be EFS, which includes second tumors, frequently registered in AYAs affected by HL [58,59].

There were fewer SMNs registered in the LH2004 than in MH’96. Even though the median observation time in LH2004 was 4 years shorter, SMNs occurrence had already reached the median time of the MH’96 protocol. The reduction of radiation treatment in terms of doses and fields could be responsible for this improvement, but it is well known that a plateau for solid tumors is not reached even at 30 years from treatment [60,61,62]. Very bizarre is the occurrence of 5 cases of PMLBCLs after a confirmed diagnosis of HL, even if secondary NHL are rising as SMNs, which were markedly higher in patients treated for HL [63] and the most frequent subtype of secondary NHL is diffuse large B-cell lymphoma (DLBCL) which has been reported in 78% of cases [64].

A different relapse time of patients younger than 15 years has emerged in our study. In LH2004, children developed relapses after a mean time of 17.9 months, which is earlier than children in MH’96 (22.3 months). They also showed an only slightly shorter median time of relapse. The use of more sensible imaging techniques in the recent protocol, like CT or PET-CT, might justify an earlier detection of relapses, but it does not happen in adolescents, whose relapse interval was superimposable. Another hypothesis of the cause of the earlier relapses could be the change of the disease in children, as the different distribution of the histologic subtypes would suggest, but it should be confirmed in the future studies with large patient populations recruited over a long period.

## 5. Conclusions

In Italy, as in other countries, the enrollment of adolescents in pediatric protocols increased along with referral to pediatric centers during the last decade. Specific training for the staff, increased attention for late effects like fertility impairment and secondary cancers, and the creation of specific “Teen areas”, (which already exist in some centers) could improve the quality of treatment of this particular group of patients. In our experience, adopting pediatric protocols did not compromise the adolescents’ outcome, as other international groups have demonstrated. Extending the pediatric oncologists’ philosophy of treatment, which entails reducing the amount of therapy and modeling strategy wherever possible to avoid long-term side effects, to adolescents may be the main road to having a population of long-term survivors with a good quality of life. Remarkable aspects of our study are the reduction of both the mixed cellularity variant and the remission period in children, as well as the reduction of low stages in both categories of age. It may be argued that these phenomena could be due to differences in the disease over time or to modification of the staging system and histologic analysis.

## 6. Strengths and Limitations

This study offers a complete overview of the HL adolescent patients treated with a national pediatric protocol in Italy, as the protocols include all adolescents aged 15–18 years old. The greatest limitation of this study is the unavailability of data to compare adolescents treated with adult protocols.

## Figures and Tables

**Figure 1 cancers-12-01620-f001:**
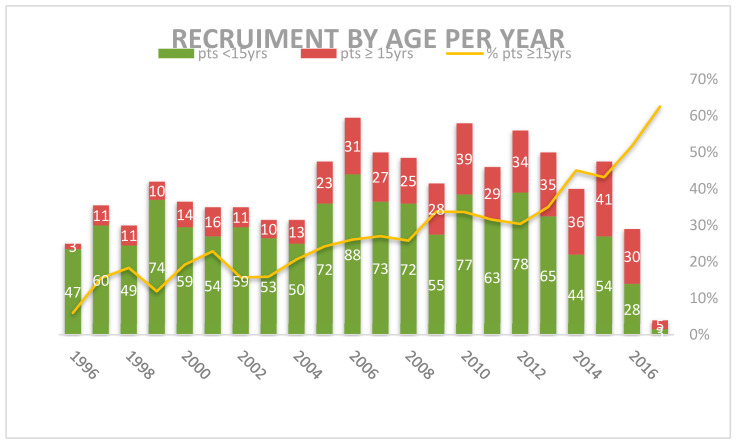
Distribution of patients by group of age per year of recruitment.

**Figure 2 cancers-12-01620-f002:**
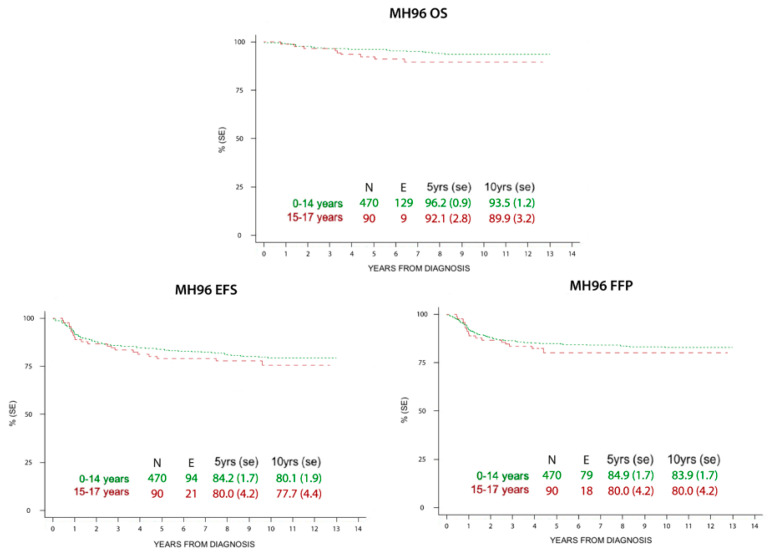
Overall survival (OS), event free survival (EFS), and freedom from progression (FFP) at 5 and 10 years in children and adolescents treated with MH’96 protocol.

**Figure 3 cancers-12-01620-f003:**
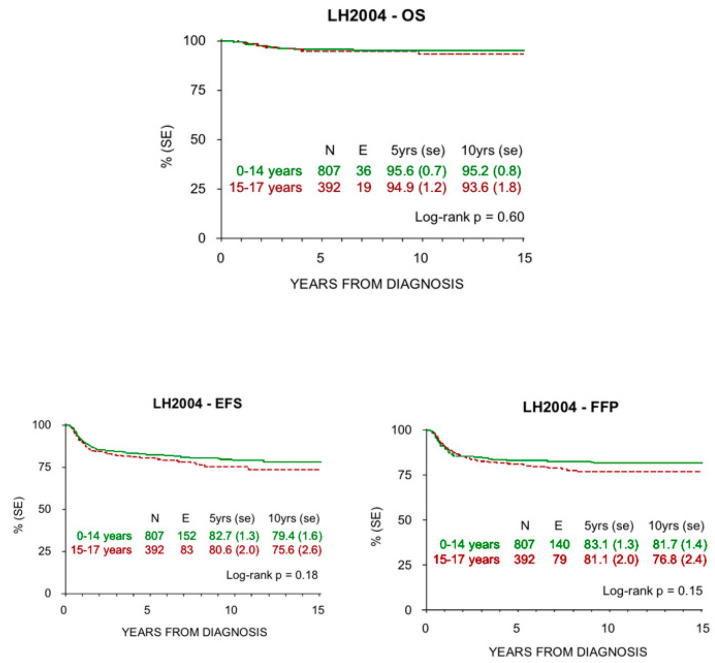
OS, EFS, and FFP at 5 and 10 years in children and adolescents treated with LH2004 protocol.

**Table 1 cancers-12-01620-t001:** MH’96 and LH2004 protocols.

Treatment Groups	MH’96	LH2004
**1** IA, IIA supradiaphragmatic no bulky no bulky, no pulmonary hilum, <4 lymphatic sites, or IA, IIA infradiaphragmatic < 4 lymphatic sites	3× ABVD +: -CR + no initial mediastinal involvement: stop -others: RT -CR o PR ≥ 75%: 20 Gy IF -PR < 75%: 36 Gy IF	3× ABVD +: -CR: stop -others: RT 25.2 Gy IF
**2** patients not included in groups 1 or 3	4× COPP/ABV + RT: -CR or PR ≥ 75%: 20 Gy IF -PR < 75%: 36 Gy IF	4× COPP/ABV +: -CR: RT 14.4 Gy IF -PR: 2× IEP + RT -CR: 14.4 Gy IF -PR: 25.2 Gy IF
**3** IIIB, IV; M/T ≥ 0.33 all stages	6× COPP/ABV + RT: -CR or PR ≥ 75%: 20 Gy IF -PR < 75%: 36 Gy IF	4× COPP/ABV +: -CR: 2× COPP/ABV + RT 14.4 Gy IF -PR: 2× IEP + RT -CR: 14.4 Gy IF -PR: 25.2 Gy IF
**If RP ≤ 50% after 2° Cycle: GR1: IEP/OPPA/COPP/IEP + RT; GR2 & 3: IEP/OPPA/IEP/OPPA/IEP + RT**		
**ABVD** Doxorubicin: 25 mg/m^2^ IV days 1 Bleomycin: 10 mg/m^2^ IV days 1 & 15 Vinblastine: 6 mg/m^2^ IV days 1 &.15 DTIC: 375 mg/m^2^ IV days 1 & 15	**COPP/ABV** Cyclophosphamide: 600 mg/m^2^ IV day 1 Vincristine: 1.4 mg/m^2^ IV day 1 Prednisone: 40 mg/m^2^ orally days 1–14 Procarbazine: 100 mg/m^2^ orally days 1–7 Doxorubicin: 35 mg/m^2^ IV day 8 Bleomycin: 10 mg/m^2^ IV day 8 Vinblastine: 6 mg/m^2^ IV day 8	**IEP** Ifosfamide: 2000 mg/m^2^ IV days 1–5 Etoposide: 120 mg/m^2^ IV days 1–5 Prednisone: 100 mg/m^2^ orally days 1–5
**OPPA** Vincristine: 1.5 mg/m^2^ IV days 1& 8 & 15 Procarbazine: 100 mg/m^2^ orally days 1–14 Prednisone: 60 mg/m^2^ IV days 1–14 Doxorubicin: 40 mg/m^2^ IV days 1&15	**COPP** Cyclophosphamide: 500 mg/m^2^ IV days 1 & 8 Vincristine: 1.5 mg/m^2^ IV days 1 & 8 Procarbazine: 100 mg/m^2^ orally days 1–15 Prednisone: 40 mg/m^2^ orally days 1–15	

CR—complete response, IF—involved fields, M/T—mediastinal-thoracic ratio, PR—partial response, RT—radiotherapy

**Table 2 cancers-12-01620-t002:** Demographic and clinical characteristics of patients analyzed and comparison between the two protocols.

	MH-‘96			LH-2004		
	Children (<15 years) N (%)	Adolescents (15–18 years) N (%)	*p* Value	Children (<15 years) N (%)	Adolescents (15–18 years) N (%)	*p* Value
**Number**	470 (83.9)	90 (16.1)		807 (67.1)	392 (32.9)	*p* = 0.000 ^1^
**Sex**			*p* = 0.429			*p* = 0.001
**Males**	277 (59.0)	49 (54.4)		470 (58.2)	189 (48.2)	
**Females**	193 (41.0)	41 (45.6)		337 (41.8)	203 (51.8)	
**Histology**			all vs. all (except n.c.): *p* = 0.024			all vs. all (except n.c.): *p* = 0.006
**Classic**						
**Nodular Sclerosis**	304 (64.7)	72 (80.0)		589 (73)	317 (80.9)	
**Mixed Cellularity**	106 (22.6)	12 (13.3)		79 (9.8)	25 (6.4)	
**Lymphocyte-Rich**	2 (0.4)			11 (1.4)	3 (0.8)	
**Lymphocyte Depletion**	3 (0.6)			8 (1.0)	2 (0.5)	
**NOS**				26 (3.2)	14 (3.6)	
**Nodular Lymphocyte Predominance**	51 (10.9)	3 (3.3)		73 (9.0)	16 (4.0)	
**n.c.**	4 (0.8)	3 (3.3)		21 (2.6)	15 (3.8))	
**Stage**			*p* = 0.259			*p* = 0.066
**I**	62 (13.2)	8 (9.0)		45 (5.6)	10 (2.5)	
**II**	222 (47.2)	52 (57.7)		405 (50.2)	216 (55.1)	
**III**	99 (21.1)	18 (20.0)		186 (23)	81 (20.7)	
**IV**	87 (18.5)	12 (13.3)		171 (21.2)	85 (21.7)	
**Symptoms**			*p* = 0.041			*p* = 0.000
**A**	309 (65.7)	49 (54.4)		503 (62.3)	202 (51.5)	
**B**	161 (34.3)	41 (45.6)		304 (37.7)	190 (48.5)	
**Mediastinum Bulky**			*p* = 0.106			*p* = 0.022
**Yes**	202 (43.0)	47 (52.2)		351 (43.5)	198 (50.5)	
**No**	268 (57.0)	43 (47.8)		456 (56.5)	194 (49.5)	
**Extra-Nodal Involvement**			*p* = 0.428			*p* = 0.304
**Yes**	101 (21.5)	16 (17.8)		208 (25.8)	112 (28.6)	
**No**	369 (78.5)	74 (82.2)		599 (74.2)	280 (71.4)	
**Treatment Group**			*p* = 0.075			*p* = 0.000
**1**	143 (30.4)	17 (19.0)		146(18.1)	35 (8.9)	
**2**	71 (15.1)	14 (15.5)		176 (21.8)	97 (24.8)	
**3**	256 (54.5)	59 (65.5)		485 (60.1)	260 (66.3)	
**RT in Group 1**			*p* = 0.048 ^2^			*p* = 0.247
**Yes**	81 (56.6)	14 (82.4)		37 (25.3)	13 (37.1)	
**No**	60 (42.0)	3 (17.6)		95 (65.1)	21 (60)	
**Missing**	2 (1.4)			14 (9.6)	1 (2.9)	
**Time to PD/Relapses**			*p* = 0.73			*p* = 0.051
**Number**	75	17		137	78	
**Mean (Months)**	22.3	20.7		17.9	20.6	
**[Range]**	[2–128]	[6–57]		[2.5–110]	[2.6–99]	
**Median (Months)**	13.3	12.8		11.9	12.1	

^1^ Age distribution between the two protocols; ^2^ Fisher test *p* = 0.065. NOS—not otherwise specified, n.c.—not classified, PD—progression of disease.

**Table 3 cancers-12-01620-t003:** Comparison of demographic and clinical characteristics of patients between the two classes of age.

	Children (<15 years)			Adolescents (15–18 years)		
	MH-’96 N (%)	LH-2004 N (%)	*p* Value	MH-’96 N (%)	LH-2004 N (%)	*p* Value
**Number**	470 (36.8)	807 (63.2)		90 (18.6)	392 (81.4)	
**Sex**			*p* = 0.808			*p* = 0.286
**Males**	277 (59.0)	470 (58.2)		49 (54.4)	189 (48.2)	
**Females**	193 (41.0)	337 (41.8)		41 (45.6)	203 (51.8)	
**Histology**			All vs. all (except n.c.): *p* = 0.000			All vs. all (except n.c.): *p* = 0.107
**Classic**						
**Nodular Sclerosis**	304 (64.7)	589 (73)		72 (80.0)	317 (80.9)	
**Mixed Cellularity**	106 (22.6)	79 (9.8)		12 (13.3)	25 (6.4)	
**Lymphocytes Rich**	2 (0.4)	11 (1.4)			3 (0.8)	
**Lymphocyte Depletion**	3 (0.6)	8 (1.0)			2 (0.5)	
**NOS**		26 (3.2)			14 (3.6)	
**Nodular Lymphocyte Predominance**	51 (10.9)	73 (9.0)		3 (3.3)	16 (4.0)	
**n.c.**	4 (0.8)	21 (2.6)		3 (3.3)	15 (3.8)	
**Stage**			*p* = 0.000			*p* = 0.015
**I**	62 (13.2)	45 (5.6)		8 (9.0)	10 (2.5)	
**II**	222 (47.2)	405 (50.2)		52 (57.7)	216 (55.1)	
**III**	99 (21.1)	186 (23)		18 (20)	81 (20.7)	
**IV**	87 (18.5)	171 (21.2)		12 (13.3)	85 (21.7)	
**Symptoms**			*p* = 0.221			*p* = 0.618
**A**	309 (65.7)	503 (62.3)		49 (54.4)	202 (51.5)	
**B**	161 (34.3)	304 (37.7)		41 (45.6)	190 (48.5)	
**Mediastinum Bulky**			*p* = 0.858			*p* = 0.770
**Yes**	202 (43)	351 (43.5)		47 (52.2)	198 (50.5)	
**No**	268 (57)	456 (56.5)		43 (47.8)	194 (49.5)	
**Extra-nodal Involvement**			*p* = 0.085			*p* = 0.001
**Yes**	101 (21.5)	208 (25.8)		16 (17.8)	112 (28.6)	
**No**	369 (78.5)	599 (74.2)		74 (82.2)	280 (71.4)	
**Therapeutic Group**			*p* = 0.000			*p* = 0.009
**1**	143 (30.4)	146 (18.1)		17 (19.0)	35 (8.9)	
**2**	71 (15.1)	176 (21.8)		14 (15.5)	97 (24.8)	
**3**	256 (54.5)	485 (60.1)		59 (65.5)	260 (66.3)	
**Radiotherapy in Group 1**			*p* = 0.000			*p* = 0.003 *
**Yes**	81 (56.6)	37 (25.3)		14 (82.4)	13 (37.1)	
**No**	60 (42)	95 (65.1)		3 (17.6)	21 (60)	
**Missing**	2 (1.4)	14 (9.6)			1 (2.9)	
**Time to PD/Relapse**			*p* = 0.012			*p* = 0.90
**Number**	75	137		17	78	
**Mean (Months)**	22.3	17.9		20.7	20.6	
**[Range]**	[2–128]	[2.5–110]		[6–57]	[2.6–99]	
**Median (Months)**	13.3	11.9		12.8	12.1	

NOS—not otherwise specified, n.c.—not classified, PD—progression of disease; *—Test Fisher: *p* = 0.004.

**Table 4 cancers-12-01620-t004:** Overall survival in children and adolescent patients related to group of treatment and era protocol.

	GROUP 1		GROUP 2		GROUP 3	
	at 5 years	at 10 years	at 5 years	at 10 years	at 5 years	at 10 years
MH96						
<15 years	98.6%	97.8%	98.6%	98.0%	94.1%	89.7%
≥15 years	100%	100%	84.6%	84.6%	91.5%	88.0%
LH2004						
<15 years	100%	100%	98.2%	97.0%	93.3%	93.0%
≥15 years	100%	100%	97.6%	97.6%	93.2%	91.2%

**Table 5 cancers-12-01620-t005:** Outcome of adolescents compared with children and adults in international studies.

Study	Country	Years	Patients N°	Treatment Protocol Pediatric/Adult, N	5 y %OS	10 y %OS	%EFS
***Significant Worse Prognosis in Adolescents vs. Children or Adults or no Improvement over Time***							
Weiner, 1997 [51]	US	1987–1992	Children (0–13 y) 76 Adolescents (14–20 y) 103	Pediatric POG 8725	-	-	5 y: 89 5 y: 72
Clavel, 2006 [21]	Europe	1978–1997	Children (0–14 y) 3528 Adolescents (15–19 y) 1862	-	1978–82: 87, 1983–87: 91, 1988–92: 93, 1993–97: 93 1978–82: 80, 1983–87: 81, 1988–92: 90, 1993–97: 88	-	-
Herbertson, 2008 [37]	UK	1969–1998	Adolescents (15–19 y) 63 Young Adults (20–25 y) 82	Adult	85.4 91.4	20 y: 76.3 20 y: 86.9	5 y: 59.9, 20 y: 56.1 5 y: 69.7, 20 y: 54.6
Keegan, 2016 [65]	US	2002–2006	Children (0–14 y) 234 AYAs (15–39 y) 2651	-	96.2 94.6		
Karim-Kos, 2016 [66]	Austria	1994–2008	Children (0–14 y) 144 Adolescents (15–19 y) 244	-	1994–98: 95.9, 1999–03: 94.7, 2004–08:100 1994–98: 97, 1999–03: 94.6, 2004–08: 91.7	-	-
Marcos-Gragera, 2018 [24]	Spain	1983–2007	Children (0–14 y) 327 Adolescents (15–19 y) 454	-	1991–95: 99, 1996–00: 86, 2001–05: 96 1991–95: 93, 1996–00: 91, 2001–05: 93	-	-
***Significant Better Prognosis in Adolescents Treated in Pediatric Centers vs. Adult Centers***							
Müller, 2011 [42]	Hungary	1990–2004	Adolescents (14–18 y) 155	Adult, 48 Pediatric, 107	89.4 92.8	83.1 89.6	5 y: 69.6, 10 y: 59.1 5 y: 82.4, 10 y: 82.4
Henderson, 2018 [43]	US	1996–2009	AYAs (17–21 y) 505	Adult E2496, 114 Pediatric COG AHOD0031, 391	89 97	-	-
***No Statistical Difference in Survival between Adolescents vs. Children or Pediatric vs. Adult Treatment***							
Cramer&Andieu, 1985 [46]	France	1972–1980	Children (0–14 y) 32 Adolescents (15–19 y) 40	Adult H 72 and H 77	-	12 y: 90.6 12 y: 92.4	-
Schellong, 1999 [47]	Germany	1990–1995	<10 155 10–15 279 15–18 136	Pediatric DAL-HD-90	97 98 97	-	92 91 92
Landman-Parker, 2000 [20]	France	1990–1996	Children (0–18 y) 202	Pediatric MDH90	97.5	-	5 y: 91.1
Mauz-Körholz, 2010 [48]	Germany	2002–2005	Children (0–12 y) 169 Adolescents (13–18 y) 404	Pediatric GPOH-HD-2002	98.2 97	-	5 y: 93.95 y: 86.9
Stark, 2015 [67]	UK	2001–2005	Children (0–12 y) 226 AYAs (13–124 y) 1494	-	95 94	-	-
Bigenwald, 2017 [25]	France	1979–2013	Adolescents (15–21 y) 176 Young Adults (21–25 y) 173	Adult	>90 >90	>90 >90	-
Englund, 2017 [22]	Denmark and Sweden	1990–2010	Children (0–9 y) 55 Adolescents (10–17 y) 364 Young Adults (18–24 y) 653	Pediatric, 315 Adult, 757	-	D 93, S 95 D 95, S 95	10 y: D 79, S 88 10 y: D 85, S 88
Fernández, 2017 [23]	US	1996–2001	Children (0–14 y) 268 Adolescents (15–21 y) 203	Pediatric P9425 and P9426	P9425: 95.4, P9426: 97.4 P9425: 92.9, P9426: 97.9	-	5 y: 87.1 5 y: 85.9
Dony, 2019 [50]	Rhône-Alpes	2000–2005	AYAs (13–25 y) 198	Pediatric, 49 Adult, 149	98 95	-	-
Raze, 2020 [49]	France	2000–2016	Adolescents (15–19 y) 557 Young Adults (20–24 y) 703	-	2000–07: 97.6, 2008–15: 98 2000–07: 95.8, 2008–15: 95.5	-	-
Zawati, 2020 [52]	Tunisia	2000–2015	Adolescents (15–25 y) 29 Young Adults (25–39 y) 37	Adult	84% 71%	-	5 y: 48% 5 y: 54%
Gupta, 2020 [45]	Canada	1992–2012	AYAs (15–21 y) 954	Pediatric, 243 Adult, 711	>90 >90	>90 >90	10 y: 83.8 10 y: 82.8
Burnelli, 2020	Italy	1996–2004	Children (0–14 y) 470 Adolescents (15–18) 90	Pediatric AIEOP MH’96	96.2 92.1	93.5 89.9	5 y: 84.2, 10 y: 80.1 5 y: 80.0 10 y: 77.7
Burnelli, 2020	Italy	2004–2017	Children (0–14 y) 807 Adolescents (15–18) 392	Pediatric AIEOP LH2004	95.6 94.9	95.2 93.6	5 y: 82.7 10 y: 79.4 5 y: 80.6 10 y: 75.6
***Significant Improvement of Adolescents’ Survival over Time***							
Koumarianou, 2007 [39]	Greece	1978–2003	AYAs (16–23 y) 55	1978–87 MOPP; 17 1988–93 MOPP/ABVD; 20 1994–03 ABVD or BEACOPP; 18	65 80 100	- - -	5 y: 53 5 y: 65 5 y: 88.5
Reedijk, 2020 [18]	Netherlands	1990–2015	Children (0–14 y) 436 Adolescents (15–17 y) 490 Young Adults (18–24 y) 1693	Pediatric and Adult	1990–94:93 to 2010–15: 98 1990–94: 84 to 2010–15: 96 1990–94: 90 to 2010–15: 98	1990–94: 80 to 2005–09: 95 1990–94: 88 to 2005–09: 94	
***Other Studies Reporting Survival Data in Adolescents***							
Yung, 2004 [36]	UK	1970–1997	Adolescents (15–17 y) 209	Adult	81	76, 20 y: 68	5 y: 50, 10 y: 45, 20 y: 41
Foltz, 2006 [41]	Canada	1981–2004	Adolescents (16–21 y) 259 Young Adults (22–45 y) 890	Adult	94 95	91, 20 y: 85 89, 20 y: 81	-
Eichenauer, 2009 [40]	Germany	1988–1998	Adolescents (15–20 y) 557 Young Adults (21–45 y) 3228	Adult GHSG HD4–9	6 y: 93.6 6 y: 90.9	12 y: 92.3 12 y: 87.1	-

AYAs—adolescents and young adults, D—Denmark, S—Sweden, y—years

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
