# Peer review of "Comparison of Hodgkin’s Lymphoma in Children and Adolescents. A Twenty Year Experience with MH’96 and LH2004 AIEOP (Italian Association of Pediatric Hematology and Oncology) Protocols"

_cancers, 2020, doi:10.3390/cancers12061620_

Round 1
Reviewer 1 Report
In this manuscript Burnelli, Mascarin on behalf the AIEOP report 20 years experience in 2 pediatric HL protocols (MH’96 and LH2004) that both enrolled hundreds of patients. There is analysis of the different relative representation of subgroups across the 2 trials and association of clinical features and outcomes). There a number of associations pointed out and there is a quite good cohort of AYA HL patients described across the analysis. This as a longitudinal data set is reasonable to publish given relative robustness of the data
Major comments:
It is not clear how this paper expands on what has been published for these trials’ past experience/outcomes. Data from MH’96 was previously published (Leuk Lymphoma. 2018 Nov;59(11):2612-2621) fairly recently. Mention of this analysis does not exist in this submitted manuscript to Cancers thus I am not sure how the current paper benefits from containing the MH’96 trial data. The only publication I could locate for the LH2004 trial (Eur J Nucl Med Mol Imaging. 2019 Jan;46(1):97-106) is focused on the subset of patients that received PET CT. Again the previous publication on LH2004 is not mentioned in this manuscript and should be referenced with a better explanation of how the outcomes from MH’96 are helpful to the paper.
In reference to comments above I think an analysis of the long term toxicity outcomes between the 2 study experiences would be useful and needed here. Also I wonder if there is potential analysis of relative outcomes between the 2 trials as there were differences between the 2 protocols namely less radiation use and lower dose radiation in LH2004. I would also be interested in analysis of disease detection in patients that got PET CT versus CT staging/response assessments.
Minor comments:
- Figure 4. It appears the color key for survival curves are transposed between the text and plotted line as there is less survival in the 15-17 year group but the curve does not reflect this.
- The abbreviations used in the Tables are very confusing. Would recommend simplifying or having better lists of a key for the abbreviations.
- It appears there was a cohort of lymphocyte predominant HL in this experience. I think this should be analyzed separated and taken out of any analysis focusing on classic HL.
- The manuscript will benefit from English language mechanics/grammar editing for readibility.
- Additional descriptions or analysis of toxicities beyond described above will help the manuscript be relevant.
Author Response
Major comments:
It is not clear how this paper expands on what has been published for these trials’ past experience/outcomes. Data from MH’96 was previously published (Leuk Lymphoma. 2018 Nov;59(11):2612-2621) fairly recently. Mention of this analysis does not exist in this submitted manuscript
This has now been included (reference n. 10)
to Cancers thus I am not sure how the current paper benefits from containing the MH’96 trial data.
The only publication I could locate for the LH2004 trial (Eur J Nucl Med Mol Imaging. 2019 Jan;46(1):97-106) is focused on the subset of patients that received PET CT. Again the previous publication on LH2004 is not mentioned in this manuscript
It was mentioned in our submitted paper, reference 34-Lopci
and should be referenced with a better explanation of how the outcomes from MH’96 are helpful to the paper.
The previous publication, regarding the results of the AIEOP MH’96, did not include an analysis on children’s and adolescents’ outcome. We decided to do that and to make a comparison with the more recent protocol, AIEOP LH2004, to evaluate if there had been any improvement. . It is well known that generally the main results of a study are published initially, and only later are other sub-analyses published, but we had the evaluation of adolescents’ characteristics and outcome ready and decided to submit it for publication. In the meantime the “main publication” is in progress and are thinking of submitting it to Cancers in the near future
In reference to comments above I think an analysis of the long term toxicity outcomes between the 2 study experiences would be useful and needed here. SNMs have been added; a complete analysis on long-term toxicity was not possible but it will be object of evaluation in the main publication on LH2004 results (Page 13 Rows 1031-1038).
Also I wonder if there is potential analysis of relative outcomes between the 2 trials as there were differences between the 2 protocols namely less radiation use and lower dose radiation in LH2004. As written before, in this analysis we considered only SNMs.
I would also be interested in analysis of disease detection in patients that got PET CT versus CT staging/response assessments. It is an interesting topic, but it is out of the realm of this analysis, which compared children and adolescents treated in different periods of time. It will be considered in the future, with an extensive evaluation of the whole patient population.
Minor comments:
- Figure 4. It appears the color key for survival curves are transposed between the text and plotted line as there is less survival in the 15-17 year group but the curve does not reflect this.
This has been corrected
- The abbreviations used in the Tables are very confusing. Would recommend simplifying or having better lists of a key for the abbreviations.
Almost all of the abbreviations have been eliminated. Where they remain, they are reported at the bottom of the table
- It appears there was a cohort of lymphocyte predominant HL in this experience. I think this should be analyzed separated and taken out of any analysis focusing on classic HL.
We have added some results about LP. In 1996 and in 2004 in Italy LP patients were treated as cHL. Only in 2009 was a specific protocol (EuroNet-PHL-LP1) opened. As is shown, there was no difference in the outcome of LP vs cHL. (pag 11, rows 513-516)
- The manuscript will benefit from English language mechanics/grammar editing for readability (sic). An American native speaker and language professor has reviewed the text.
- Additional descriptions or analysis of toxicities beyond described above will help the manuscript be relevant. As reported before, the analysis of SNMs has been added (Pag 11, rows 517-527, Page 13 Rows 1031-1038).
Reviewer 2 Report
This is an extensive restrospective analysis of Hodgkin's lymphoma in children and adolescents; however, I think that this segment of patients might only be of limited interest to the readers of Cancers; but this is for the Editorial Board to decide.
One of my main points of criticism is that while the authors have analyzed outcomes, this especially vulnerable group in terms of long term side effects (i.e. fertility, secondary malignancies, cardiac diseases...) is not seen in light of outcome vs toxicities, as the authors make no real mention of this. For potential publication, I think that this information should be improved, as the data-set should allow for extracting these important results!
Author Response
This is an extensive restrospective analysis of Hodgkin's lymphoma in children and adolescents; however, I think that this segment of patients might only be of limited interest to the readers of Cancers; but this is for the Editorial Board to decide.
One of my main points of criticism is that while the authors have analyzed outcomes, this especially vulnerable group in terms of long term side effects (i.e. fertility, secondary malignancies, cardiac diseases...) is not seen in light of outcome vs toxicities, as the authors make no real mention of this. For potential publication, I think that this information should be improved, as the data-set should allow for extracting these important results! SNMs have been added. A complete analysis on long-term toxicity was not possible but it will be object of evaluation in the main publication on LH2004 results (Pag 11, rows 517-527, Page 13 Rows 1031-1038).
Reviewer 3 Report
This is well written retropsective analysis of two large Italian HL studies comparing the results across children and adolescents. The title should be amended to reflect that both children and adolescents have been analysed ( title suggests it is restricted to adolescents).
Some other minor suggestions to improve the manuscript are
Results
To improve readability some of the tables ( such as Table 4 and 5) can be included as supplementary material
Discussion:
For e.g a good reason to detect early relapse in the LH2004 trial may be due to better use of imaging such as PET scans - this should be discussed rather than attributing to an unknown biological reason
The discussion and comparisons with various AYA trials is lengthy and difficult to follow. I suggest summarising in a table and shortening the discussion for this section.
Changes in the nos of histological subtypes are also likely due to the panel of immunohistochemistry available and changing over time rather than just biopsy issues.
Author Response
Comments and Suggestions for Authors
This is well written retropsective analysis of two large Italian HL studies comparing the results across children and adolescents. The title should be amended to reflect that both children and adolescents have been analysed ( title suggests it is restricted to adolescents). The title has been modified
Some other minor suggestions to improve the manuscript are
Results
To improve readability some of the tables ( such as Table 4 and 5) can be included as supplementary material Tables 4 and 5 have been eliminated because they regarded the LH2004 protocol exclusively.
Discussion:
For e.g a good reason to detect early relapse in the LH2004 trial may be due to better use of imaging such as PET scans - this should be discussed rather than attributing to an unknown biological reason. A discussion about the use of PET-CT has been added(Page 13 rows 1041-1043)
The discussion and comparisons with various AYA trials is lengthy and difficult to follow. I suggest summarising in a table and shortening the discussion for this section. Table 6 has been included, reporting studies about children and AYA.
Changes in the nos of histological subtypes are also likely due to the panel of immunohistochemistry available and changing over time rather than just biopsy issues. This aspect has been discussed (Page 12 Rows 684-688)
Round 2
Reviewer 2 Report
Secondary malignancies have been included now, which strengthens the manuscript and show that they are mostly related to radiation.